# Prevalence and intensity of soil-transmitted helminth infections in Narok and Bomet Counties, Kenya: Evidence from program monitoring

Paul Kibati [1,2]*, Florence Wakesho[2], Stephen Mwatha[1,2], Denver Mariga[1,2], Josephine Ihahi[1,2], Dickson Kioko[2], Joseph Otieno Oloo[2], Elvis Oyugi[1], Sammy M. Njenga[3], Ben Masiira[4], Maurice Owiny[1,4], Cara Tupps[5], Mariana Stephens[5], Paul Emerson[5], Wyckliff Omondi[2], Rubina Imtiaz[5], Sultani Matendechero[6], Sanjaya Dhakal[5], Kristin M. Sullivan[5]

**1** Field Epidemiology and Laboratory Training Program, Ministry of Health, Nairobi, Kenya, **2** Division of Vector-Borne and Neglected Tropical Diseases, Ministry of Health, Nairobi, Kenya, **3** Kenya Medical Research Institute, Nairobi, Kenya, **4** African Field Epidemiology Network, Kampala, Uganda, **5** Children Without Worms, The Task Force for Global Health, Decatur, Georgia, United States of America, **6** National Public Health Institute, Nairobi, Kenya

* pgkibati@gmail.com

**Editor:** jong-Yil Chai, Seoul National University College of Medicine, KOREA, REPUBLIC OF

## Abstract

### Background

The backbone of Kenya's soil-transmitted helminthiasis control program is the periodic distribution of anti-helminthic preventive chemotherapy (PC). PC distribution has been conducted since 2012 through the National School-Based Deworming Programme, which deworms school-age children (SAC, 5–14 years) living in areas at high risk for soil-transmitted helminth (STH) infections. After nearly a decade of deworming, the Ministry of Health sought to generate evidence for program monitoring by estimating the prevalence and intensity of STH infection in Narok and Bomet counties among at-risk groups: preschool-aged children (PSAC, 2–4 years), SAC, and women of reproductive age (WRA, 15–49 years), of which only SAC are covered by the deworming program.

### Methods

During August and December of 2021, we conducted cross-sectional, population-based household surveys in Narok and Bomet counties, using multi-stage, cluster random sampling among resident PSAC, SAC, and WRA. Individual and household questionnaires were administered using an electronic mobile platform. Stool samples were collected and tested for roundworms (*Ascaris lumbricoides*), hookworms (*Necator americanus*, *Ancylostoma duodenale*), and whipworms (*Trichuris trichura*) using the Kato-Katz method.

**Data availability statement:** The data used for the analyses in this paper are owned by the government of Kenya, and are subject to provisions of the Data Protection Act https://kenyalaw.org/kl/fileadmin/pdfdown-loads/LegalNotices/2021/LN263_2021.pdf. Researchers are welcome to request access to de-identified datasets from the Division of Vector-Borne and Neglected Tropical Diseases at the Ministry of Health (headdvbntd@gmail.com).

**Funding:** This research was funded by a grant provided to Children Without Worms (Georgia, USA) by Johnson and Johnson to support surveys conducted by the Division of Vector-Borne and Neglected Tropical Diseases, Ministry of Health, Kenya. The funders had no role in the survey design, data collection, data analysis, or interpretation. The findings and conclusions of the survey reflect the view of the authors only" as indicated on the manuscript.

**Competing interests:** I have read the journal's policy and the authors of the manuscript have the following competing interests: KMS receives a salary from The Task Force for Global Health, an organization that receives funding from GSK and Johnson & Johnson, the manufacturers of albendazole and mebendazole, respectively.

## Results

Stool samples were provided by 1,062 PSAC, 1,922 SAC, and 364 WRA. Results indicated that the prevalence of any STH infection in Bomet county was similar among SAC (16.2% [upper 95% design-corrected confidence interval [U95% CI]: 23.1%]) and PSAC (15.8% [U95% CI: 21.7%]). In Narok county, STH prevalence was marginally higher among PSAC (12.8% [U95% CI: 18.2%]) compared to SAC (11.3% [U95% CI: 16.2%]). Moderate-to-high intensity infection prevalence among PSAC and SAC exceeded the morbidity elimination threshold of 2% in both counties. *A. lumbricoides* and *T. trichura* were most identified; few hookworm infections were detected.

## Conclusions

STH infections remain a public health problem in Narok and Bomet counties. There may be a need to expand PC distribution to include risk groups beyond SAC during deworming exercises, as envisioned in the Breaking Transmission Strategy, to hasten progress toward the achievement of STH as a public health problem.

### Author summary

Soil-transmitted helminth (STH) infections, including roundworms (*Ascaris lumbricoides*), hookworms (*Necator americanus, Ancylostoma duodenale*), and whipworms (*Trichuris trichura*), have been shown to affect children's nutrition and cognitive development. Monitoring the prevalence of STH infections among at-risk populations remains a key step in determining the type and frequency of deworming medication to administer. Globally, treatment guidelines recommend treating children and women of reproductive age (WRA), as they are at the highest risk of morbidity from infection. In Kenya, preventive chemotherapy (PC) administration has primarily targeted school-aged children (SAC) using a school-based delivery platform. However, program monitoring surveys demonstrated that the prevalence and intensity of STH infections among children continues to exceed the country's elimination targets. Our results showed that in Narok and Bomet counties, the prevalence of any STH infection was moderate (10–<20%) among children in both counties and low (2–<10%) among WRA. This suggests the need to consider expanding PC administration to a community-based model to make significant strides toward eliminating soil transmitted helminths as a public health problem.

### Introduction

Soil-transmitted helminth (STH) infections caused by roundworms (*Ascaris lumbricoides*), hookworms (*Necator americanus*, *Ancylostoma duodenale*), and whipworms

(*Trichuris trichura*) are a public health problem globally, predominantly in the poorest regions of the world [1,2]. The World Health Organization (WHO) classifies soil-transmitted helminthiasis among the neglected tropical diseases (NTDs) and it is estimated that STH affects more than 1.5 billion people globally [3]. In 2021, it was estimated that there were more than 640 million STH cases, resulting in more than 3400 deaths and 1.38 million disability-adjusted life years [4].

STHs cause human infection through the ingestion of eggs from contaminated food and soil or by active skin penetration in the case of hookworms [5]. STH infections are known to cause both acute and chronic disease conditions such as anaemia, malnutrition, abdominal pain and discomfort, diarrhoea, as well as impaired physical and cognitive development [5,6]. WHO recommends controlling morbidity caused by STH infections through preventive chemotherapy (PC) with the anthelmintic drugs albendazole or mebendazole [7].

In Kenya, STH infections are widely distributed, occurring mainly in western and southeastern coastal regions, as well as other localized areas [5,8,9]. According to Kenya's 2016–2020 NTD strategic plan [10], it is estimated that 10 million people nationwide are affected by STH, with children accounting for 50%. Another 16.6 million people are deemed to be at risk of contracting STH infections [10–12]. The Ministry of Health, through the Division of Vector-Borne and Neglected Tropical Diseases, launched the Breaking Transmission Strategy in 2019, which aimed to eliminate morbidity due to STH infections [13]. This strategy seeks to integrate interventions by focusing on increasing coverage of mass drug administration (MDA), establishing sustainable implementation of necessary NTD-related water, sanitation, and hygiene interventions, and implementing behaviour change communication interventions that are necessary to move from control to elimination of STH infections as a public health problem [14].

The launch of the Breaking Transmission Strategy was necessitated by the fact that since 2012, PC has been provided by the National School-Based Deworming Programme targeting school-aged children (SAC, 5–14 years) in various parts of the country [15]. Within the first three years of implementation, the program demonstrated a reduction of moderate-to-heavy intensity (MHI) infection [15]. However, it was noted that STH reinfections remained common following treatment, suggesting the need for expanded interventions [16,17] and that, in areas of high prevalence, adults and children not in school acted as reservoirs of STH, thereby impeding the effectiveness of STH control programs through school-based interventions only [18,19].

Given this concern, in 2021, the Ministry of Health Kenya, through the Division of Vector-Borne and Neglected Tropical Diseases, sought to better understand the community-wide prevalence of infection. They selected two counties, Narok and Bomet, where there had been minimal reduction in STH prevalence among SAC, despite ongoing school-based deworming efforts spanning over five years [16]. This survey aimed to provide programmatic evidence supporting community-wide deworming by determining the point prevalence and intensity of STH infections among PSAC, SAC, and WRA in Narok and Bomet Counties [20–22].

## Objective

The objective of these community-based, cross-sectional surveys was to determine the prevalence and intensity of STH infections among preschool-aged children (PSAC, ages 2–4 years), SAC (ages 5–14 years), and women of reproductive age (WRA, 15–49 years) in Narok County and Bomet Counties.

## Methods

### Ethics statement

Ethical approval for the surveys was granted by the Ethics and Scientific Review Committee of AMREF Health Africa in Kenya (P 705 – 2019). Permission to collect data was obtained from the respective county/sub-county health offices. Participants and guardians were provided with information about the purpose of the survey and what to expect. Written informed consent by parents or guardians and verbal assent by participating children was obtained before enrolment.

While this survey did not provide immediate treatment post-sample collection, all individuals and guardians of participating children received comprehensive health education regarding STH transmission, prevention and treatment availability through local health services.

## Survey setting and population

The surveys were conducted in Narok and Bomet counties (Fig 1) in August and December 2021, respectively. These counties were purposefully selected due to their varying ecological and epidemiological contexts, historical PC administration through the National School Based Deworming Program from 2012, and ongoing concerns about persistent transmission of STH.

Narok County covers an area of 17,933 km$^2$ with a population of roughly 1.2 million. The county is administratively divided into six sub-counties and 30 wards, each serving as smaller administrative sub-units of a sub-county [13].

Bomet County has an area of 2,037 km$^2$ with a projected population of roughly 900,000. The county is divided into five sub-counties and 25 wards [23].

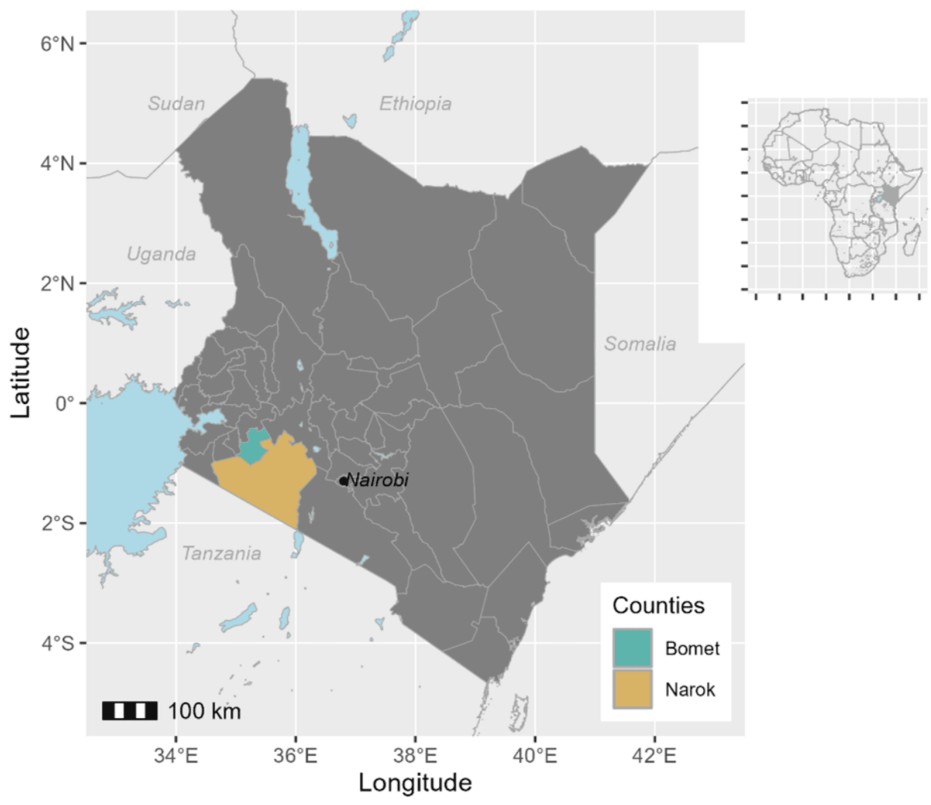

**Fig 1. Map of Kenya showing the locations of Bomet and Narok counties.** County and national boundaries were derived from openly licensed shapefiles provided by the United Nations Office for the Coordination of Humanitarian Affairs (OCHA) via the Humanitarian Data Exchange (HDX) platform (https://data.humdata.org/dataset/cod-ab-ken). Map created using R version 4.3.2, with ggplot2 on 28APR2022. **Basemap Information:** Kenya - Subnational Administrative Boundaries (https://data.humdata.org/dataset/cod-ab-ken); License: Creative Commons Attribution for Intergovernmental Organisations.

## Survey design

We conducted a community-based, cross-sectional survey using a multi-stage cluster sampling approach following the Integrated Community-based Survey for Program Monitoring methodology. Similar sampling designs have been implemented in similar epidemiological studies in Uganda, Sierra Leone and Bangladesh [24–26]. Full methodological details can be found in the online reference manual (https://childrenwithoutworms.org/icspm-reference-manual/), with key details provided below.

The population-based, household surveys were conducted using a multi-stage, cluster random sampling technique, producing equal selection probabilities among resident PSAC, SAC, and WRA in the respective counties. Within each county, population and housing data from the Kenya National Bureau of Statistics, supplemented by county administrators, were used to identify the villages, which served as clusters [20]. Thirty clusters were randomly selected from each county in the first stage using population proportional to estimated size. The team selected clusters using the Survey Sample Builder developed by the Neglected Tropical Disease Support Centre at the Task Force for Global Health (https://childrenwithoutworms.org/icspm-survey-sample-builder-tool/).

The sample size for this survey was based on the transmission assessment survey methodology published by WHO, which recommends at least 332 participants for each at-risk group in a cluster survey [27,28]. Sample size calculation targeted the lowest precision adequate for assessing whether the prevalence of any STH infection had fallen below WHO thresholds for public health intervention [29,30].

To ensure the sample size was reached, 475 participants were selected for inclusion for each risk group in each county to accommodate an approximate non-response of 30%. Non-response was anticipated to cover both participants who did not consent to be surveyed and those who did not provide stool samples. Households were systematically selected using a pre-determined sampling interval, and the household selection form aided field teams in knowing which risk group was to be sampled from the household.

## Recruitment and sample collection

Household members of selected households were included if they: a) were a member of the selected risk groups of interest (PSAC, SAC, and WRA), b) slept at the household the night before the interview, c) planned to stay at the household the night of the interview, d) were able to provide a single stool sample within 24 hours of the interview, and e) were able and willing to provide verbal consent or assent.

On the survey day, field officers visited the selected households, and explained the purpose of the survey and ensured consent/assent was obtained. The field teams were comprised of a national-level supervisor from the Ministry of Health, a county-level supervisor, a community health assistant (CHA) from the area, and a community health volunteer (CHV) from the village being visited. Each participant was assigned a unique identification number, which was used to track the collected sample and link it to the individual information collected during the household visit. Participants were interviewed by trained data collectors using a pre-tested, structured questionnaire. Socio-demographic information on the household was provided by the household head/designate, and everyone provided their individual-level information. Information for young children was provided by the household head/designate.

The participants were given a labelled wide-mouthed plastic container for stool collection, a piece of plain paper, a piece of applicator sticks, and other sanitary necessities such as tissue paper. The plastic containers were labelled with the participants' unique identification numbers and their name, age, and sex to ensure a mix-up of containers did not occur within the household. The survey information was given to the head of household or his designate if the household head was absent. If the household head was not conversant in English, the community health assistant explained the instructions in the local language. The field team explained safe stool collection to the participant and asked them to wash their hands with soap and water after providing samples. The fresh stool samples were collected the morning after the household visit and interview.

## Laboratory procedure

Stool samples were transported to the laboratory using cooler boxes within three hours of collection. To address the recognized sensitivity limitations of the Kato-Kats technique, particularly for hookworm detection due to rapid egg degradation in low transmission settings, we established mobile laboratories at the nearest health centre to minimize specimen transport time [27]. Stool samples were then prepared using a 41.7 mg Kato-Katz template, with duplicate slides prepared and read by two laboratory technicians under microscopes [22]. To maintain quality assurance, 10% of the slides were randomly selected and examined by a third technician blinded from the initial results from the other technicians.

## Classification of infection intensity

The infection intensity, commonly called worm burden, is typically measured indirectly by counting the number of eggs in faeces, expressed as eggs per gram (epg). This is because indirect methods are more convenient and less invasive [24]. Direct methods of classifying worm burden, which involve counting the number of expelled worms after anti-helminthic treatment, may not be feasible during mass surveys and in areas with low worm burden.

WHO classifies infection intensity as light, moderate, or heavy using indirect methods for each type of STH [24]. The epg estimate was applied to categorize the samples based on infection intensity following WHO guidelines. For *A. lumbricoides* infections, light intensity was defined as 1–4,999 epg, moderate intensity as 5,000–49,999 epg, and heavy intensity as ≥50,000 epg. *T. trichiura* infections were classified as light intensity with 1–999 epg, moderate intensity with 1,000–9,999 epg, and heavy intensity with ≥10,000 epg. For hookworm infections, light intensity was defined as 1–1,999 epg, moderate intensity as 2,000–3,999 epg, and heavy intensity as ≥4,000 epg.

## Data collection and management

Data were collected via mobile phone using Open Data Kit [ODK] and uploaded to a central secure server. Three data sets [individual, household, and laboratory] were downloaded from the server and saved on computers at the Ministry of Health Kenya. Using Microsoft Excel, the data were cleaned, and the household and individual files were merged with unique household identification numbers and later with the laboratory file's unique individual identification numbers.

## Data analysis

The cleaned dataset was imported into SAS version 9.04 and R version 4.13 to manage and analyze the data.. In line with the ICSPM survey methodology, the point prevalence and intensity of STH infections were estimated separately for each parasite species and stratified by county and risk group. We reported one-sided upper 95% confidence intervals (U95%CI) as the survey was designed to determine the lowest threshold under which true prevalence lies.

## Results

### Characteristics of survey population

The survey teams visited 60 villages/clusters in the two counties, with 30 surveyed in each Bomet and Narok. However, due to an electronic data collection issue, participant information from one village in Narok County was lost. A total of 3,348 (48% male, 51% female and 1% unreported) individuals were enrolled in the surveys, of whom 2,335 (69.7%) provided stool samples (Table 1).

Among the 2,096 children providing a stool sample, 1,034 (49.3%) were female, and the median age was 3 years for PSAC (interquartile range [IQR]:2–4), 9 years for SAC (IQR:7–12), and 29 years for WRA (IQR: 20–40).

PLOS Neglected Tropical Diseases

**Table 1. Number of clusters, households, and individuals surveyed by country and risk group, Bomet and Narok Counties, 2021.**

| County | Clusters* | | Households | | Individuals | | | |
|---|---|---|---|---|---|---|---|---|
| | Sur-veyed | House-holds per cluster | En-rolled | Enrolled households where ≥ 1 member provided a stool sample | Risk group | En-rolled | Among those enrolled, number of individuals providing stool sample | Among those providing stool sample, number female |
| | n | mean (SD) | n | n (%) | | n | n (%) | n (%) |
| Bomet | 30 | 21.8 (6.0) | 654 | 530 (81.0) | PSAC | 508 | 361 (71.1) | 186 (51.5) |
| | | | | | SAC | 1,014 | 749 (73.9) | 387 (51.7) |
| | | | | | WRA | 225 | 148 (65.8) | |
| Narok | 29† | 18.9 (6.3) | 547 | 429 (78.4) | PSAC | 554 | 367 (66.2) | 186 (50.1) |
| | | | | | SAC | 908 | 619 (68.2) | 275 (44.4) |
| | | | | | WRA | 139 | 91 (65.5) | |

Abbreviations: n–Count; %-prevalence; PSAC–preschool-aged children (2–4 years old); SAC–school-aged children (5–14 years old); SD–Standard Deviation; WRA–women of reproductive age (15–49 years old).

*Clusters are typically a single village; however, in some cases, villages are combined if the projected village size in households is less than the target segment size.

†Information from surveys in one cluster was lost due to an electronic data collection issue. Therefore, only information on 29 clusters was included in the analysis.

## Prevalence of infection

The prevalence of infections appeared to be slightly higher in Bomet County than in Narok County across all risk groups (Table 2). Among children (PSAC and SAC), the estimated prevalences ranged from 11% to 13% in Narok County and approximately 16% in Bomet County. Among WRA, in both counties, the prevalence of any STH infection was less than 10%, with Narok at 6.6% and Bomet at 8.1%.

**Table 2. Estimated prevalence of soil-transmitted helminth infection by county, species, intensity, and risk group, Bomet and Narok Counties, 2021.**

| County | Risk group | Providing stool samples | Hookworm* | *A. lumbricoides* | *T. trichiura* | Any STH |
|---|---|---|---|---|---|---|
| | | n | % (95% UCL) | % (95% UCL) | % (95% UCL) | % (95% UCL) |
| **Any intensity infections** | | | | | | |
| Bomet | PSAC | 361 | 0.0 | 10.8 (16.1) | 6.4 (10.3) | 15.8 (21.7) |
| | SAC | 749 | 0.0 | 11.3 (15.9) | 8.4(15.2) | 16.2 (23.1) |
| | WRA | 148 | 0.0 | 6.1 (11.9) | 4.1 (9.4) | 8.1 (14.5) |
| Narok | PSAC | 367 | 0.3 (1.2) | 9.5 (14.5) | 4.6 (7.2) | 12.8 (18.2) |
| | SAC | 619 | 0.3 (1.5) | 6.3 (10.0) | 6.8(10.7) | 11.3 (16.2) |
| | WRA | 91 | 1.1 (5.1) | 4.4 (10.8) | 2.2 (6.7) | 6.6 (13.3) |
| **Moderate-to-heavy intensity infections** | | | | | | |
| Bomet | PSAC | 361 | 0.0 | 5.8 (9.3) | 0.8 (2.1) | 6.1 (9.5) |
| | SAC | 749 | 0.0 | 5.5 (8.7) | 0.1 (0.6) | 5.5 (8.7) |
| | WRA | 148 | 0.0 | 0.7 (3.2) | 0.0 | 0.7 (3.2) |
| Narok | PSAC | 367 | 0.0 | 2.7 (5.7) | 1.1 (2.9) | 3.3 (6.3) |
| | SAC | 619 | 0.0 | 2.4 (4.3) | 0.5 (1.6) | 2.7 (4.7) |
| | WRA | 91 | 0.0 | 3.3 (8.0) | 0.0 | 3.3 (8.0) |

Abbreviations: n–Count; %–Prevalence; UCL–Upper 95% one-sided confidence limit; PSAC–preschool-aged children (2–4 years old); SAC–school-aged children (5–14 years old); STH–Soil-transmitted helminth; WRA–women of reproductive age (15–49 years old).

*N. americanus and A. duodenale.

In both counties, the prevalence of MHI infection among children was above the 2% threshold for elimination as a public health problem. Children in Bomet County appeared to have higher MHI infection prevalence among both PSAC (6.1%) and SAC (5.5%) than corresponding risk groups in Narok County. WRA in Bomet County recorded the lowest prevalence (0.7%) among all risk groups.

## Parasite type

In both counties, *A. lumbricoides* caused the most infections, followed by *T. trichura.* Hookworm infections were not detected in Bomet County and were uncommon in Narok County. In Bomet County, the prevalence of *A. lumbricoides* infections was slightly higher among SAC, followed by PSAC, then WRA. In Narok County, *T. trichura* infections followed this pattern; however, unlike Bomet County, the prevalence of *A. lumbridcoides* was shown to be higher among PSAC than SAC.

## Co-infections

Co-infections were uncommon (Table 3). When identified, they were most frequently *A.lumbricoides* and *T.trichura,* with all risk groups in both counties recording at least one co-infection. No triple infections were detected.

## Discussion

Our findings underscore persistent transmission of STH, evidenced by moderate (10–<20%) STH infection prevalence among children in Bomet and Narok Counties. Prevalence estimates of any STH infection were above the 2% treatment threshold target according to the breaking transmission strategy [31], and the prevalence of MHI infections also exceeded the WHO 2030 target for elimination as a public health problem (<2%) among all surveyed children [7]. These findings are consistent with recent findings from geostatistical modelling of STH using a nationwide survey dataset that showed a high probability of sub counties in the southwestern part of Kenya where Narok and Bomet are located exceeding the 2% and 10% prevalence threshold for *A.lumbricoides* and *T.trichura* [32].

These results are critical for informing programmatic decisions, particularly on the continuation of MDAs. While the 2% threshold is an established benchmark for guiding such interventions, its interpretation must consider local transmission dynamics [33,34]. Similar patterns of persistent infection despite continued PC among SAC have been highlighted in other

**Table 3. Estimated soil-transmitted helminth coinfection prevalence by county and risk group, Bomet and Narok Counties, 2021.**

| County | Risk group | Providing stool samples | Uninfected | Mono-infection | | | Co-infection | | | Triple infection |
|---|---|---|---|---|---|---|---|---|---|---|
| | | | | Hookworm* | *A. lumbricoides* | *T. trichiura* | Hookworm and *A. lumbricoides* | Hookworm and *T. trichiura* | *A. lumbricoides* and *T. trichiura* | Hookworm, *A. lumbricoides,* and *T. trichiura* |
| | | n | n (%) | n (%) | n (%) | n (%) | n (%) | n (%) | n (%) | n (%) |
| Bomet | PSAC | 361 | 304 (84.2) | 0 (0.0) | 34 (9.4) | 18 (5) | 0 (0.0) | 0 (0.0) | 5 (1.4) | 0 (0.0) |
| | SAC | 749 | 628 (83.8) | 0 (0.0) | 58 (7.7) | 36 (4.8) | 0 (0.0) | 0 (0.0) | 27 (3.6) | 0 (0.0) |
| | WRA | 148 | 136 (91.9) | 0 (0.0) | 6 (4.1) | 3 (2.0) | 0 (0.0) | 0 (0.0) | 3 (2.0) | 0 (0.0) |
| Narok | PSAC | 367 | 320 (87.2) | 0 (0.0) | 29 (7.9) | 12 (3.3) | 1 (0.3) | 0 (0.0) | 5 (1.4) | 0 (0.0) |
| | SAC | 619 | 549 (88.7) | 1 (0.2) | 27 (4.4) | 29 (4.7) | 0 (0.0) | 1 (0.2) | 12 (1.9) | 0 (0.0) |
| | WRA | 91 | 85 (93.4) | 1 (1.1) | 3 (3.3) | 1 (1.1) | 0 (0.0) | 0 (0.0) | 1 (1.1) | 0 (0.0) |

Abbreviations: n–Count; p–Prevalence; PSAC–preschool-aged children (1–4 years old); SAC–school-aged children (5–14 years old); WRA–women of reproductive age (15–49 years old).

*N. americanus and A. duodenale.

studies, reflecting the relative slow decline of infections despite continued school based deworming [35–37]. These findings suggest that tailoring of continued annual PC in these counties is required to make progress toward the elimination of STH morbidity as a public health problem in the country.

Compared to prior surveys, the estimated prevalences among SAC from our community-based surveys (Bomet, 16.2%, and Narok, 11.3%) were lower than those demonstrated by school-based impact surveys conducted in 2017, five years after the commencement of the National School-Based Deworming Programme. These 2017 surveys showed a reduction of STH among SAC in Bomet from 29.7% in 2012 to 18.1% in 2017 and, in Narok, from 53% in 2012 to 41% in 2017 [16]. However, the prevalences are higher than those derived from the ten-year model based geostatistical impact surveys (Bomet 8.3%, and Narok 10.8%) conducted between 2021 and 2022 [25]. Despite there being an overall decline in the prevalence of STH from baseline in the country, the rate of decline in these two counties has been relatively lower, which could be explained by poor sanitation facilities, resulting in possible re-infection [26].

## Parasite species

*A. lumbricoides* was the most common STH infection observed in the two counties, followed by *T. trichura,* with very few hookworm infections detected. This finding corresponds with school-based surveys conducted by the National School-based Deworming Programme, which found *A. lumbricoides* to be the most common STH during baseline, midline, and end-line surveys in the country as well as geostatistical modelling that showed *A.lumbricoides* and *T.trichura* to be prevalent in these areas [18,32]. This may be due to helminths' long lifespan in human hosts and their production of many environmentally resistant eggs, which are expelled in faeces to enhance their survival and lifecycle continuation, leading to reinfection [17,38].

The survey also concurs with findings from the National School-based Deworming Programme that did not show significant reductions in the prevalence of MHI infection with *T. trichura,* where albendazole is the drug distributed. In contrast, it has been shown that a combination of ivermectin and albendazole would be the drug of choice in areas with high *T. trichura* infections [26,39]*,* and WHO now suggests including ivermectin in addition to albendazole in such settings [40,41]. Drugs such as emodepside are also currently being tested in humans and show great promise as alternative treatments to address the limited effectiveness of benzimidazoles against *T. trichiura* [42]*.*

It is also possible that reinfection occurs among the at-risk age groups, resulting in a resurgence of prevalence, as stated in a meta-analysis that showed infection after treatment occurs rapidly, particularly for *A. lumbricoides* and *T.trichura* [17].

## Risk groups targeted by PC

Both infection prevalence and MHI infection prevalence were similar among PSAC and SAC within both counties. The consistency in high prevalence among PSACs who are routinely excluded from PC distributions highlights the fact that they should be considered for targeting during deworming exercises, as infections among this age group may lead to morbidity and cognitive impairment and may compromise the performance of PSAC on expressive language [38].

In addition, given the prevalence of similar magnitude among PSAC and SAC, the Ministry may consider expanding beyond school-based PC to include other at-risk groups, such as PSAC and WRA, to achieve broader community impact, as shown in studies from western and coastal Kenya and Malawi [32,41,43,44]. This expansion beyond SAC may enable the country to accelerate progress toward eliminating STH as a public health problem in line with the Breaking Transmission Strategy and the WHO 2030 NTD Road Map, similar to the approach taken by Kenya to conduct community wide mapping in the coastal part of Kenya, coupled with higher coverage rates and the ability of reaching non-enrolled SAC through community wide MDAs [9,45]. While Kenya has implemented community wide MDAs in select parts of the country, additional evaluation of its cost effectiveness relative to other delivery models could guide programmatic decisions to determine the most feasible strategy for progressing towards elimination.

## Limitations

The survey was designed to estimate prevalence based on county-level administrative boundaries. However, intra-county prevalence heterogeneity is likely. While subgroup comparative analyses were not conducted for this study, as the ICSPM methodology prioritizes population-level classification rather than inferential analyses, this could be explored in future work. Hence, the program was not able to consider the diversity of the ecological conditions that have implications on the drivers of STH transmission within the counties, contributing to the heterogeneous classification of infection. In addition, the study did not assess individual, behavioural or environmental risk factors. This may limit interpretation of the observed intra-county heterogeneity limiting insights into drivers of transmission. This, in turn, would affect decision-making as to where PC distribution would be most efficient and cost-effective for the program. Subsequently, with these results, PC administration would be conducted in the entire county and not in specific focal areas with a prevalence above 2%, contrary to the BTS, which stipulates that the implementation unit for treatment is the ward.

The sensitivity of the Kato Katz technique in detecting hookworm eggs is reduced in low-prevalence settings and with sample transportation delays between the time of collection in the field and analysis in the laboratory. Even under optimal conditions, its diagnostic sensitivity remains low for hookworm, with studies showing that this method may substantially underestimate true infection prevalence [46–48]. This ought to be considered when interpreting programmatic interpretation of prevalence estimates. However, to mitigate this risk, laboratories were established at the nearest health centres where teams were conducting the survey. This significantly reduced the time between sample collection and laboratory analysis. Delays in sample shipment would lead to an underestimation of the true prevalence of hookworm infections in the sampled population [27,28]. In addition, some participants who were provided with stool pots didn't provide samples. However, the number was not significant as the sample size for the study was still achieved.

During the survey conducted in Narok County, data from one village was lost, reducing the sample size. This loss may have influenced the STH prevalence estimate in the county. However, it's anticipated that any shifts in the county-level prevalence would likely be minimal given that data from the missing village represent approximately 1/30th of the total sample size.

## Conclusion

Our survey showed that STH infections remain a public health problem in Narok and Bomet counties. The elevated infection prevalence and MHI infection prevalence among all surveyed groups, including PSAC and WRA, suggest that expansion to risk groups beyond SAC during deworming exercises, as envisioned in the Breaking Transmission Strategy, may accelerate progress to controlling STH infections.

## Acknowledgments

We thank the children and parents who participated in this survey, and the supervision and technical support from the Division of Vector-Borne and Neglected Tropical Diseases, Children Without Worms, and the African Field Epidemiology Network [AFENET]. We are grateful to the field officers who collected the data and the data team who worked tirelessly to ensure the survey was completed. We acknowledge the support from the national and county management in Bomet and Narok counties.

## Author contributions

**Conceptualization:** Mariana Stephens, Paul Emerson, Rubina Imtiaz, Sultani Matendechero.

**Data curation:** Paul Kibati.

**Formal analysis:** Paul Kibati, Kristin M. Sullivan.

**Funding acquisition:** Mariana Stephens, Sultani Matendechero.

**Investigation:** Paul Kibati, Florence Wakesho, Stephen Mwatha, Denver Mariga, Dickson Kioko, Joseph Otieno Oloo, Ben Masiira, Cara Tupps, Wyckliff Omondi, Sultani Matendechero.

**Methodology:** Florence Wakesho, Elvis Oyugi, Mariana Stephens, Paul Emerson, Rubina Imtiaz, Sultani Matendechero, Sanjaya Dhakal.

**Project administration:** Paul Kibati, Florence Wakesho, Dickson Kioko, Joseph Otieno Oloo, Elvis Oyugi, Ben Masiira, Maurice Owiny, Cara Tupps, Mariana Stephens, Paul Emerson, Wyckliff Omondi, Rubina Imtiaz, Sultani Matendechero, Sanjaya Dhakal, Kristin M. Sullivan.

**Resources:** Paul Kibati, Florence Wakesho, Dickson Kioko, Cara Tupps, Mariana Stephens, Paul Emerson, Wyckliff Omondi, Rubina Imtiaz, Sultani Matendechero.

**Software:** Paul Kibati, Kristin M. Sullivan.

**Supervision:** Paul Kibati, Florence Wakesho, Dickson Kioko, Joseph Otieno Oloo, Ben Masiira, Cara Tupps, Wyckliff Omondi, Sultani Matendechero.

**Validation:** Paul Kibati.

**Visualization:** Paul Kibati, Kristin M. Sullivan.

**Writing – original draft:** Paul Kibati.

**Writing – review & editing:** Paul Kibati, Florence Wakesho, Stephen Mwatha, Denver Mariga, Josephine Ihahi, Dickson Kioko, Joseph Otieno Oloo, Elvis Oyugi, Sammy M. Njenga, Ben Masiira, Maurice Owiny, Cara Tupps, Mariana Stephens, Paul Emerson, Wyckliff Omondi, Rubina Imtiaz, Sultani Matendechero, Sanjaya Dhakal, Kristin M. Sullivan.

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
