## [Decision Letter · Decision Letter 0]

25 Feb 2025

Response to Reviewers
Revised Manuscript with Track Changes
Manuscript

Kind regards,

Shaden Kamhawi

co-Editor-in-Chief

Paul Brindley

co-Editor-in-Chief

**Journal Requirements:**

At this stage, the following Authors/Authors require contributions: Paul Kibati, Florence Wakesho, Stephen Mwatha, Denver Mariga, Josephine Ihahi, Dickson Kioko, Joseph Otieno Oloo, Elvis Oyugi, Sammy M. Njenga, Ben Masiira, Maurice Owiny, Cara Tupps, Mariana Stephens, Paul Emerson, Wyckliff Omondi, Rubina Imtiaz, Sultani Matendechero, Sanjaya Dhakal, and Kristin M. Sullivan. Please ensure that the full contributions of each author are acknowledged in the "Add/Edit/Remove Authors" section of our submission form.

3) Tables should not be uploaded as individual files. Please remove these files and include the Tables in your manuscript file as editable, cell-based objects. For more information about how to format tables, see our guidelines:

https://journals.plos.org/plosntds/s/tables

Potential Copyright Issues:

- Figure 1. Please provide a direct link to the base layer of the map (i.e., the country or region border shape) and ensure this is also included in the figure legend; and provide a link to the terms of use / license information for the base layer image or shapefile. We cannot publish proprietary or copyrighted maps (e.g. Google Maps, Mapquest) and the terms of use for your map base layer must be compatible with our CC BY 4.0 license.

6) We note that your Data Availability Statement is currently as follows: "The data used for the analyses in this paper are owned by the government of Kenya. Researchers are welcome to request access to de-identified datasets from the Division of Vector-Borne and Neglected Tropical Diseases at the Ministry of Health (Wyckliff.omondi@gmail.com) or through Children Without Worms (cww@taskforce.org).". Please confirm at this time whether or not your submission contains all raw data required to replicate the results of your study. Authors must share the “minimal data set” for their submission. PLOS defines the minimal data set to consist of the data required to replicate all study findings reported in the article, as well as related metadata and methods (https://journals.plos.org/plosone/s/data-availability#loc-minimal-data-set-definition).

- The points extracted from images for analysis..

**Reviewers' comments:**

**Key Review Criteria Required for Acceptance?**

**Methods** :

-Are the objectives of the study clearly articulated with a clear testable hypothesis stated?

-Is the study design appropriate to address the stated objectives?

-Is the population clearly described and appropriate for the hypothesis being tested?

-Is the sample size sufficient to ensure adequate power to address the hypothesis being tested?

-Were correct statistical analysis used to support conclusions?

-Are there concerns about ethical or regulatory requirements being met?

Reviewer #1: (No Response)

Reviewer #2: - The objectives of the study are clearly mentioned in an own paragraph as part of the introduction. I suggest moving this part to the methods but including an additional "aim-sentence" at the end of the introduction.

- The authors aimed to asses the current STH prevalence in two different counties in Kenya. They conducted a cross-sectional survey, which is a good tool to get a prevalence assessment at one time point.

- The authors clearly describe that the aim was to assess prevalence among PSAC, SAC and WRA as they belong to the high-risk groups. The authors conducted a cross-sectional household survey. This method ensured to include the PSAC, the WRA and SAC even though they might not go to school.

- The sample size described by the authors comes with multiple issues:

== the authors decided for a sample size of 332 based on WHO recommendations for STH prevalence assessment in the context of LF transmission assessment surveys. However, in the present manuscript, no LF transmission assessment survey was done. Hence, I have doubts that the number can just be used for solely doing an STH prevalence assessment.

== The sentence describing the number 332 and the WHO recommendations does not cite the WHO recommendations but a paper by Katz

== In line 153, it is not clear what defines a "district" as above the only units described are counties and sub-counties.

- In lines 214-216, the authors describe the calculations they conducted. I wonder why the authors decided to only provide the upper confidence interval and not the lower given that the WHO goal of 2% is rather on the lower than the upper side in this context. Furthermore, the authors present in the results different sub analyses, e.g. per species or per risk group. It would be good to mention this here, too.

- There are no concerns about ethical or regulatory requirements being met. The authors asked for written consent of adults or legal guardians of children and verbal assent of participating children. For future studies, I would recommend, however, to include a written assent of children from the age of e.g. 8 onwards as well.

Reviewer #3: • While the manuscript details the sampling process, it lacks a statistical justification for the sample size. The authors should explain whether the chosen sample size is powered to detect significant differences in prevalence between the risk population groups and counties. Additionally, discussing potential limitations due to non-response would add rigor.

• The manuscript applies the WHO infection intensity thresholds, but it does not justify their relevance to the study setting. Given the varying endemicity levels and transmission dynamics in Narok and Bomet, a brief discussion on why these thresholds are suitable would be beneficial. This 2% threshold is a global benchmark intended to guide public health interventions. However, its applicability can vary based on local endemicity and transmission dynamics. In areas with high transmission rates, reinfection can occur rapidly after treatment, potentially leading to fluctuations in infection intensity even if the prevalence temporarily falls below 2% (see Truscott et al, 2021). Conversely, in low-endemicity settings, maintaining the 2% threshold might require sustained interventions to prevent resurgence (see Werkman et al, 2018). Therefore, while the 2% threshold serves as a useful guideline, tailoring control strategies to local conditions is essential for effective and sustainable STH management.

• The manuscript acknowledges the limitations of Kato-Katz in detecting hookworm eggs, particularly in low-prevalence settings, and describes a mitigation strategy by reducing sample transport time. However, even under optimal conditions, the Kato-Katz method has inherently low sensitivity for hookworm diagnosis due to rapid egg degradation and variability in egg excretion patterns. The authors should discuss whether their approach fully mitigated this limitation and consider citing studies (e.g., Barreto et al., 2017; Nikolay et al., 2014) that show how Kato-Katz underestimates hookworm prevalence. If feasible, future studies could validate findings using more sensitive diagnostic techniques, such as PCR or FLOTAC.

• The ethics statement is well-documented, but given that deworming is a standard intervention, were any specific measures taken to provide treatment to infected individuals following sample collection? This should be mentioned.

**Results** :

-Does the analysis presented match the analysis plan?

-Are the results clearly and completely presented?

-Are the figures (Tables, Images) of sufficient quality for clarity?

Reviewer #1: (No Response)

Reviewer #2: - The analysis mainly matches the analysis plan. However, the results are more detailed that what is described in the analysis plan, i.e. the results present prevalence per e.g. species or risk group, which is not described in the methods.

- The figures and tables are nice and seem to have good quality. However, for one of the figures, I wonder why the authors decided to present it in black and white. A comment can be found in the attached word document.

Reviewer #3: • The manuscript presents prevalence estimates with upper 95% confidence limits but does not discuss how design effects were accounted for. Clarifying the method for computing confidence intervals (e.g., whether adjustments were made for clustering effects) would be helpful.

• Table 2 presents prevalence by county and risk group, but it is unclear whether these differences are statistically significant. Adding p-values or confidence intervals for comparisons would enhance interpretability.

• Figure 2 effectively visualizes infection prevalence, but it does not highlight co-infection patterns. Given that co-infections were observed, a separate figure showing co-infection frequencies might be useful.

**Conclusions** :

-Are the conclusions supported by the data presented?

-Are the limitations of analysis clearly described?

-Do the authors discuss how these data can be helpful to advance our understanding of the topic under study?

-Is public health relevance addressed?

Reviewer #1: (No Response)

Reviewer #2: - The discussion is supported by the data presented. However, the discussion could be made a bit broader, taking other studies and countries into account and draw conclusions not only from their own results but in relationship to other studies. I added two examples in the comments to the discussion part. The authors could write a bit more details about the future implications of their and other researcher's results.

- The limitations are clearly described

- The public health relevance is described

Reviewer #3: • The manuscript suggests expanding deworming to preschool-aged children (PreSAC) and women of reproductive age (WRA). However, it does not fully explore the operational challenges of such an expansion. A discussion on feasibility, cost implications, and alternative delivery strategies (e.g., integrating PC into maternal health services) would strengthen this recommendation.

• Given the persistent prevalence of moderate-to-heavy infections despite ongoing mass drug administration (MDA), it would be useful to discuss reinfection dynamics. Are there environmental or behavioural factors that could be contributing to this persistence? Linking the findings to broader discussions on water, sanitation, and hygiene (WASH) interventions could provide a more holistic interpretation.

**Editorial and Data Presentation Modifications?**

Reviewer #1: (No Response)

Reviewer #2: (No Response)

Reviewer #3: (No Response)

**Summary and General Comments** :

Reviewer #1: (No Response)

Reviewer #2: Congratulations to the authors for a nicely written manuscript and nice study. The manuscript covers the prevalence assessment of STH in particularly regions in Kenya and the survey type chosen is a good one to achieve the aim. I primarily have some suggestions and minor comments on how to improve the study even further. The major issue is the sample size calculation that, in my view, should have been based on previous prevalence knowledge of the region, and the number of individuals living there. Some of my comments are to be found here and some more are in the document attached.

Reviewer #3: This manuscript presents findings on soil-transmitted helminth (STH) infections in Narok and Bomet counties, Kenya, using a community-based survey approach. The study is valuable as it provides updated epidemiological data that could inform policy decisions regarding deworming strategies. The findings are well-supported by data and align with current public health priorities in helminth control.

However, some areas require further refinement to enhance clarity, rigor, and the impact of the findings: i) justification for methodological choices, particularly the sample selection and infection intensity thresholds, ii) a more in-depth discussion on how findings align with existing literature, and iii) addressing potential biases and limitations that may affect the interpretation of results.

Major Comments:

1. Study design and methodology:

• While the manuscript details the sampling process, it lacks a statistical justification for the sample size. The authors should explain whether the chosen sample size is powered to detect significant differences in prevalence between the risk population groups and counties. Additionally, discussing potential limitations due to non-response would add rigor.

• The manuscript applies the WHO infection intensity thresholds, but it does not justify their relevance to the study setting. Given the varying endemicity levels and transmission dynamics in Narok and Bomet, a brief discussion on why these thresholds are suitable would be beneficial. This 2% threshold is a global benchmark intended to guide public health interventions. However, its applicability can vary based on local endemicity and transmission dynamics. In areas with high transmission rates, reinfection can occur rapidly after treatment, potentially leading to fluctuations in infection intensity even if the prevalence temporarily falls below 2% (see Truscott et al, 2021). Conversely, in low-endemicity settings, maintaining the 2% threshold might require sustained interventions to prevent resurgence (see Werkman et al, 2018). Therefore, while the 2% threshold serves as a useful guideline, tailoring control strategies to local conditions is essential for effective and sustainable STH management.

• The manuscript acknowledges the limitations of Kato-Katz in detecting hookworm eggs, particularly in low-prevalence settings, and describes a mitigation strategy by reducing sample transport time. However, even under optimal conditions, the Kato-Katz method has inherently low sensitivity for hookworm diagnosis due to rapid egg degradation and variability in egg excretion patterns. The authors should discuss whether their approach fully mitigated this limitation and consider citing studies (e.g., Barreto et al., 2017; Nikolay et al., 2014) that show how Kato-Katz underestimates hookworm prevalence. If feasible, future studies could validate findings using more sensitive diagnostic techniques, such as PCR or FLOTAC.

2. Statistical Analysis and Data Presentation

• The manuscript presents prevalence estimates with upper 95% confidence limits but does not discuss how design effects were accounted for. Clarifying the method for computing confidence intervals (e.g., whether adjustments were made for clustering effects) would be helpful.

• Table 2 presents prevalence by county and risk group, but it is unclear whether these differences are statistically significant. Adding p-values or confidence intervals for comparisons would enhance interpretability.

• Figure 2 effectively visualizes infection prevalence, but it does not highlight co-infection patterns. Given that co-infections were observed, a separate figure showing co-infection frequencies might be useful.

3. Ethical considerations

• The ethics statement is well-documented, but given that deworming is a standard intervention, were any specific measures taken to provide treatment to infected individuals following sample collection? This should be mentioned.

4. Discussion:

• The manuscript suggests expanding deworming to preschool-aged children (PreSAC) and women of reproductive age (WRA). However, it does not fully explore the operational challenges of such an expansion. A discussion on feasibility, cost implications, and alternative delivery strategies (e.g., integrating PC into maternal health services) would strengthen this recommendation.

• Given the persistent prevalence of moderate-to-heavy infections despite ongoing mass drug administration (MDA), it would be useful to discuss reinfection dynamics. Are there environmental or behavioural factors that could be contributing to this persistence? Linking the findings to broader discussions on water, sanitation, and hygiene (WASH) interventions could provide a more holistic interpretation.

Minor Comments:

1. Abstract: Consider simplifying the Background to focus more on key findings and implications. In conclusions, emphasize actionable recommendations (e.g., programmatic changes needed for STH control).

2. Discussion: Clarify whether prevalence reductions observed in the study are statistically meaningful compared to previous estimates. Expand on the role of alternative treatment regimens (ie. incorporating ivermectin), particularly for Trichuris trichiura, which is known to be less responsive to albendazole.

References

1. Werkman M, Wright JE, Truscott JE, Easton AV, Oliveira RG, Toor J, Ower A, Ásbjörnsdóttir KH, Means AR, Farrell SH, Walson JL, Anderson RM. Testing for soil-transmitted helminth transmission elimination: Analysing the impact of the sensitivity of different diagnostic tools. PLoS Negl Trop Dis. 2018 Jan 18;12(1):e0006114. doi: 10.1371/journal.pntd.0006114. PMID: 29346366; PMCID: PMC5773090.

2. Truscott JE, Hardwick RJ, Werkman M, Saravanakumar PK, Manuel M, Ajjampur SSR, Ásbjörnsdóttir KH, Khumbo K, Witek-McManus S, Simwanza J, Cottrell G, Houngbégnon P, Ibikounlé M, Walson JL, Anderson RM. Forecasting the effectiveness of the DeWorm3 trial in interrupting the transmission of soil-transmitted helminths in three study sites in Benin, India and Malawi. Parasit Vectors. 2021 Jan 20;14(1):67. doi: 10.1186/s13071-020-04572-7. PMID: 33472677; PMCID: PMC7818558.

3. Barreto, R. E., Narváez, J., Sepúlveda, N. A., Velásquez, F. C., Bello, S. D., López, M. C., Reyes, P., & Moncada, L. I. (2017). Combination of five diagnostic tests to estimate the prevalence of hookworm infection among school-aged children from a rural area of colombia. In Acta Tropica (Vol. 173, p. 160). Elsevier BV. https://doi.org/10.1016/j.actatropica.2017.05.028

4. Nikolay B, Brooker SJ, Pullan RL. Sensitivity of diagnostic tests for human soil-transmitted helminth infections: a meta-analysis in the absence of a true gold standard. Int J Parasitol. 2014 Oct 1;44(11):765-74. doi: 10.1016/j.ijpara.2014.05.009. Epub 2014 Jun 30. PMID: 24992655; PMCID: PMC4186778.

PLOS authors have the option to publish the peer review history of their article (what does this mean? ). If published, this will include your full peer review and any attached files.

**Do you want your identity to be public for this peer review?** For information about this choice, including consent withdrawal, please see our Privacy Policy .

Reviewer #1: **Yes: ** Gaoussou COULIBALY

Reviewer #2: No

Reviewer #3: No

**Figure resubmission:****Reproducibility:** To enhance the reproducibility of your results, we recommend that authors of applicable studies deposit laboratory protocols in protocols.io, where a protocol can be assigned its own identifier (DOI) such that it can be cited independently in the future. Additionally, PLOS ONE offers an option to publish peer-reviewed clinical study protocols. Read more information on sharing protocols at https://plos.org/protocols?utm_medium=editorial-email&utm_source=authorletters&utm_campaign=protocols

---

## [Decision Letter · Decision Letter 1]

15 Jun 2025

* A rebuttal letter that responds to each point raised by the editor and reviewer(s). You should upload this letter as a separate file labeled 'Response to Reviewers '. This file does not need to include responses to any formatting updates and technical items listed in the 'Journal Requirements' section below.

Revised Manuscript with Track Changes
Manuscript

Section Editor

Shaden Kamhawi

co-Editor-in-Chief

Paul Brindley

co-Editor-in-Chief

**Comments to the Authors:**

**Please note that two reviews are uploaded as attachments.**

**Reviewers' comments:**

**Key Review Criteria Required for Acceptance?**

**Methods**

-Are the objectives of the study clearly articulated with a clear testable hypothesis stated?

-Is the study design appropriate to address the stated objectives?

-Is the population clearly described and appropriate for the hypothesis being tested?

-Is the sample size sufficient to ensure adequate power to address the hypothesis being tested?

-Were correct statistical analysis used to support conclusions?

-Are there concerns about ethical or regulatory requirements being met?

Reviewer #1: i) Are the objectives of the study clearly articulated with a clear testable hypothesis stated?

Acceptable, Even if the link with the hypothesis is not very clear.

ii) Is the study design appropriate to address the stated objectives?

Yes, But some shortcomings in applicability.

iii) Is the population clearly described and appropriate for the hypothesis being tested?

Yes

iv) Is the sample size sufficient to ensure adequate power to address the hypothesis being tested?

approximately

v) Were correct statistical analysis used to support conclusions?

No statistical tests were actually carried out, just descriptive data. It would be necessary to compare the groups (PSAC, SAC, WRA) and according to demographic parameters (sex, age, etc.).

vi) Are there concerns about ethical or regulatory requirements being met?

I would have preferred ethical approval by the National Ethics Committee. Given that this is a national issue.

Reviewer #2: The methods read more detailed now than last time. I am a bit confused that they are placed at the end of the manuscript, squeezed in between required information for the journal. I think this should be solved and the methods should be placed before the results. Made it a bit hard to jump back and forth between methods and results, too.

Reviewer #3: The revised manuscript addresses the main concerns raised in the initial review. The study objectives are clearly stated and appropriate for the public health surveillance context. The cross-sectional design is suitable, and the target population is well defined.

Sample size justification is now aligned with ICSPM and WHO guidelines, and the statistical methods used (one-sided U95% CIs, DEFF = 2.0) are appropriate for threshold-based programmatic decisions. Ethical procedures are adequately described, including consent, health education, and referrals for treatment.

Overall, the methodology is sound, the data are well analyzed, and the revisions improve the manuscript’s clarity and relevance.

**Results**

-Does the analysis presented match the analysis plan?

-Are the results clearly and completely presented?

-Are the figures (Tables, Images) of sufficient quality for clarity?

Reviewer #1: i) Does the analysis presented match the analysis plan?

Acceptable with some shortcomings.

ii) Are the results clearly and completely presented?

Some aspects are missing...see section v of methods. Intra- and extra-group stratification analyses would be necessary. These results could be presented in an appendix, for example!

iii) Are the figures (Tables, Images) of sufficient quality for clarity?

Please match the places and dates in the titles of the figures and tables. Figure legends are not marked. Figure numbers do not appear! North is not shown on the map.

Reviewer #2: The results read fine and are presented in tables and figures. Here some comments on the results:

- I think it would be nice to include a flowchart to show the recruitment process in more detail and include i) in how many households was nobody home or no adult/head of household, ii) in how many households, individuals did not want to consent, iii) how many individuals consented but did not provide a stool sample, iv) how many individuals were excluded because they did not fulfil the eligibility criteria such as staying in the household the night before and after.

- I think a logistic regression would have been a nice add-on to this publication. I think it is not a must but would give the presentation more substance.

- The caption of Figure 2 should include the abbreviation used for the legend of the Figure. I am aware that the heading in the figure itself includes it but for completeness I would include it in the caption or even consider to exchange the HMI abbreviation for the full wording in the figure.

Reviewer #3: The analysis presented is consistent with the ICSPM methodology and aligns with the stated objectives and analysis plan. The results are clearly and comprehensively presented, including stratification by species, risk group, and infection intensity. Figures and tables are of good quality and effectively support the findings. The use of Table 3 to present co-infection patterns is appropriate and sufficiently detailed.

**Conclusions**

-Are the conclusions supported by the data presented?

-Are the limitations of analysis clearly described?

-Do the authors discuss how these data can be helpful to advance our understanding of the topic under study?

-Is public health relevance addressed?

Reviewer #1: i) Are the conclusions supported by the data presented?

Yes

ii) Are the limitations of analysis clearly described?

Not really!

iii) Do the authors discuss how these data can be helpful to advance our understanding of the topic under study?

Yes, but a few more sentences would be necessary!

v) Is public health relevance addressed?

You could put it this way. Authors should highlight the importance of the study to the public and the scientific community.

Reviewer #2: The discussion needs a bit more discussion.

Two examples:

1. The authors write: "Despite there being an overall decline in the prevalence of STH from baseline in the country, the rate of decline in these two counties has been relatively lower, which could be explained by poor sanitation facilities, resulting in possible re-infection[26]." -- So the prevalence declined but not as much and you give poor sanitation facilities as the reason. I would suggest to discuss how these could be improved. Were they already improved a little bit, therefore the decline? Where does the decline come from if its not the sanitation facilities? What would the counties need to improve them, of who is this the responsibility? Also, how did other countries manage to bring the prevalence down? Was it also because of WASH facilities or are there any other ways to reduce prevalence?

2. The authors write: "In contrast, it has been shown that a combination of ivermectin and albendazole would be the drug of choice in areas with high T. trichura infections[26]. Additionally, it has been shown that there is heterogeneity of STH transmission, which would necessitate a different approach in areas found to have high T. trichuris infections.(41). It is also possible that reinfection occurs among the at-risk age groups, resulting in a resurgence of prevalence, as stated in a meta-analysis that showed infection after treatment occurs rapidly, particularly for A. lumbricoides and T.trichura [16]." -- This paragraph reads incomplete to me. You write about a drug combination but you do not discuss how this might impact the prevalence in your setting. You write about heterogeneity and a different approach but you do not mention the approach or what this would mean for your counties or elsewhere. And you write about reinfections but you do not discuss how they could be prevented.

Reviewer #3: The conclusions are well supported by the data presented. The manuscript clearly describes limitations, including diagnostic sensitivity and geographic heterogeneity. The discussion effectively contextualizes findings within broader regional experiences and outlines implications for expanding preventive chemotherapy. The public health relevance is well articulated, particularly regarding policy decisions to reach underserved risk groups and sustain progress toward STH elimination targets.

**Editorial and Data Presentation Modifications?**

Reviewer #1: Authors should review the order of the writing plan! The "Methods" section appears after the acknowledgments... I don't know if this is a requirement of the editorial line!

Reviewer #2: lines 76-79: Reference number 3 is from 2010 and therefore rather old. And reference number 4 is referring to a study solely conducted in Ethiopia. It does not seem to be a good fit for the sentence here. What about the WHO weekly epidemiological report from 2024?

line 88 and many others: There is no consistency in presenting the references. Sometimes it is a squared bracket, somethings its round brackets and in this case here its both. I would recommend to unify them and then check if they are all still correct or if because of the different referencing methods, there has been a mix of references.

line 119: Are there multiple Bomet Counties?

line 149: Exchange to lumbricoides

line 319: Here, your inclusion criteria is verbal assent or consent. But further up in your ethics, you write about written consent. Which one is true?

Reviewer #3: The revised manuscript is well written and clearly presented. Figures and tables are appropriate and support the findings effectively. Minor editorial suggestions include ensuring clarity in a few expanded discussion points—particularly around the feasibility and cost implications of expanding PC delivery to preschool-aged children and women of reproductive age. While the authors cite relevant pilot experiences, a more explicit discussion of implementation challenges (e.g., health system integration, community delivery platforms) would further strengthen the operational relevance of the recommendations.

These are minor suggestions that do not affect the overall conclusions.

**Summary and General Comments**

Reviewer #1: Methods

Ethical Consideration

Wouldn't there be a National Ethics Committee in Kenya for ethical approval?

Response: Ethical approval was granted by the Amref Health Africa Ethics and Scientific Review Committee, which is accredited by the National Commission for Science, Technology and Innovation (NACOSTI). At the time of the study, this was deemed sufficient to conduct the study

Reviewe#1: From now on, you have to go through the National Ethics Committee in addition to the institutional commission.

Were children aged 2-4 able to give verbal consent? What was the benefit for participants, particularly those diagnosed as positive?

Response: For children aged 2-4 years, verbal assent was provided by the parent/caregiver. While no deworming was provided immediately post-survey, participants received health education and were referred to ongoing local PC programs. This approach is consistent with the survey protocol and ethical approvals.

Reviewe#1: This statement comes from you “Written informed consent by parents or guardians and verbal assent by participating children was obtained before enrolment”.

Survey setting and population

Why did you choose particularly Narok and Bomet for this study? Is there an explanation?

What were the criteria used to select the composite locations?

Response: We appreciate this observation and query by the reviewer regarding the selection of these counties. These counties were purposively selected due to persistent STH transmission despite multiple rounds of preventive chemotherapy. The composite locations were defined as clusters and sampling as per the ICPSM protocol to ensure representativeness across the three at-risk-groups.

Reviewe#1: Please include this information in the manuscript. Also, make the “hotspot” aspect part of the title.

The methods section is poorly structured!

We appreciate the reviewer’s helpful feedback regarding the structure of the Methods section. In response, we have reorganised and expanded the section to follow a clearer and more logical structure, directly aligned with the ICSPM survey methodology.

The revised Methods section now includes the following distinct subheadings:

1. Ethics Statement

2. Survey Setting and Population

3. Survey Design

4. Recruitment and Sample Collection

5. Laboratory Procedure

6. Classification of Infection Intensity

7. Data Collection and Management

8. Data Analysis

Reviewe#1: No major changes in the structure!

Statistical analysis

The statistical analysis section is not rich in information. No statistical tests have been applied! I suggest that you carry out comparative analyses by gender, age group (intra county) and between counties.

Reviewe#1: No response was given to this request! This is crucial!!!

What did you do with the questionnaire data? Wasn't it possible to highlight risk factors?

Response: We thank the reviewer for the comments. As per the ICSPM methodology we did not assess the risk factors in this manuscript. However, the questionnaire data were collected for future risk factor analysis, which was beyond the scope of this baseline prevalence manuscript required for programmatic decision making on PC administration.

Reviewe#1: We can't make any programmatic decisions without taking stock of the risk factors... that would be putting the cart before the horse! If the data is there, why not exploit it?

Reviewer #2: I think the authors have targeted an important topic, and its important to know the local prevalences of Counties for STH. However, I think the discussion needs a bit more discussion, including what the results mean for the implementation of interventions for the area and how they could be implemented. What is needed to reduce the prevalence in the area and how can this be solved? What does the WHO say and how did other NTD programmes or studies successfully reduced prevalence? What are the implications of your study?

Reviewer #3: Major comments

• Justification of Sample Size and Design Effect: The authors have now provided sufficient justification for the sample size based on ICSPM (Integrated Community-based Survey for Program Monitoring) and WHO TAS guidelines, explaining that the sample allows for classifying prevalence thresholds with appropriate power. They also clarified the assumed design effect (DEFF = 2.0), consistent with ICSPM methodology. This addresses previous concerns.

• Applicability of WHO’s 2% Threshold: The revised manuscript includes a more nuanced discussion about the relevance of the 2% MHI threshold, referencing literature (e.g., Truscott et al., 2021; Werkman et al., 2018) to explain how transmission dynamics may affect interpretation. This addition strengthens the discussion and aligns the study with programmatic and contextual considerations.

• Kato-Katz Limitations and Diagnostic Sensitivity: The manuscript now adequately acknowledges that even with mitigation measures (use of local/mobile laboratories), Kato-Katz likely underestimates hookworm prevalence in low-endemicity settings. The proposed references were appropriately included. Future use of more sensitive diagnostics (e.g., PCR) is acknowledged.

• Operational Feasibility of Expanding PC to PreSAC and WRA: The authors have expanded the discussion by referencing pilot experiences in Kenya and Malawi.

• Consideration of Reinfection and WASH: The revised discussion addresses how poor sanitation and limited access to WASH contribute to persistent transmission and reinfection. This strengthens the public health interpretation and links prevalence findings to broader structural issues.

• Interpretation of Prevalence Changes Over Time: While the manuscript notes that the survey was not powered to detect statistically significant changes from earlier surveys, it now clarifies that MHI prevalence remains above elimination thresholds, justifying the need for continued MDA.

• Alternative Treatments for Trichuris trichiura: The revised manuscript includes a brief but important mention of ivermectin-albendazole combination therapy, citing recent studies. This addresses concerns about the limited efficacy of albendazole for T. trichiura.

Minor Comments

• The authors clarified that one-sided upper confidence intervals were used to assess if prevalence is below thresholds, following ICSPM protocol. While two-sided intervals or p-values might enhance comparative interpretation, the current approach aligns with the survey’s programmatic objectives.

• The revised ethics section now specifies that participants received health education and referrals to existing programs. This addresses the concern regarding post-survey follow-up in the absence of on-site treatment.

• The abstract has been revised to emphasize the main findings and recommendations. The conclusions are now more actionable, suggesting programmatic expansion of PC to underserved groups.

• Although a separate figure for co-infections was not included, the data are clearly presented in Table 3. This is acceptable and consistent with the structure of the manuscript.

PLOS authors have the option to publish the peer review history of their article (what does this mean? ). If published, this will include your full peer review and any attached files.

**Do you want your identity to be public for this peer review?** For information about this choice, including consent withdrawal, please see our Privacy Policy .

Reviewer #1: **Yes: ** Gaoussou Coulibaly, Université Félix Houphouët-Boigny, Abidjan, Côte d'Ivoire

Reviewer #2: No

Reviewer #3: No

**Figure resubmission:****Reproducibility:** To enhance the reproducibility of your results, we recommend that authors of applicable studies deposit laboratory protocols in protocols.io, where a protocol can be assigned its own identifier (DOI) such that it can be cited independently in the future. Additionally, PLOS ONE offers an option to publish peer-reviewed clinical study protocols. Read more information on sharing protocols at https://plos.org/protocols?utm_medium=editorial-email&utm_source=authorletters&utm_campaign=protocols

---

## [Decision Letter · Decision Letter 2]

4 Nov 2025

Response to Reviewers
Revised Manuscript with Track Changes
Manuscript

Shaden Kamhawi

co-Editor-in-Chief

Paul Brindley

co-Editor-in-Chief

**Additional Editor Comments :**

**Comments to the Authors:**

**Please note that one review is uploaded as an attachment.**

**Reviewers' comments:**

**Key Review Criteria Required for Acceptance?**

**Methods**

-Are the objectives of the study clearly articulated with a clear testable hypothesis stated?

-Is the study design appropriate to address the stated objectives?

-Is the population clearly described and appropriate for the hypothesis being tested?

-Is the sample size sufficient to ensure adequate power to address the hypothesis being tested?

-Were correct statistical analysis used to support conclusions?

-Are there concerns about ethical or regulatory requirements being met?

Reviewer #1: (No Response)

Reviewer #3: The updated version has satisfactorily addressed the methodological concerns raised in earlier reviews. The study objectives are now clearly articulated within a programmatic surveillance context, and while no formal hypothesis is tested, the aims are appropriately framed for the descriptive design used. The choice of study design has been justified and is suitable to meet the stated objectives. The target populations—preschool-aged children, school-aged children, and women of reproductive age—are clearly described and remain appropriate for assessing prevalence across risk groups. Importantly, the authors have strengthened their justification of sample size, explicitly referencing ICSPM and WHO TAS methodology and clarifying the design effect assumption (DEFF = 2.0), thereby addressing earlier concerns about statistical power. The statistical analyses also follow the ICSPM protocol, with clear rationale for the use of one-sided upper confidence intervals to determine whether prevalence falls below programmatic thresholds, which resolves the previous ambiguity on analytical approach. Ethical procedures have been clarified, with explicit mention of AMREF’s accredited approval, correction of consent/assent wording, and confirmation that participants received health education and referrals, thereby addressing earlier concerns about post-survey follow-up. Overall, the revisions have resolved the methodological issues raised in earlier rounds, and the study now presents a sound and well-justified framework to support its conclusions.

**Results**

-Does the analysis presented match the analysis plan?

-Are the results clearly and completely presented?

-Are the figures (Tables, Images) of sufficient quality for clarity?

Reviewer #1: (No Response)

Reviewer #3: In the updated version, the results section has been strengthened and addresses the concerns raised in earlier reviews. The analysis presented is consistent with the predefined ICSPM analysis plan and is clearly aligned with the programmatic objectives of classifying prevalence against WHO thresholds. The results are now more clearly and comprehensively presented, with prevalence estimates disaggregated by species, risk group, and infection intensity, and the limitations of the analyses explicitly acknowledged. Tables and figures have been revised for consistency, with corrected numbering, clearer captions, and inclusion of all relevant abbreviations, thereby improving readability. Co-infection data are appropriately summarized in Table 3, which is sufficiently detailed and makes a separate figure unnecessary.

While subgroup comparative analyses were not conducted, the justification based on ICSPM methodology is acceptable; however, a short statement acknowledging that such analyses could be explored in future work would further strengthen the manuscript.

Overall, the results are now complete, clearly presented, and supported by figures and tables of adequate quality.

**Conclusions**

-Are the conclusions supported by the data presented?

-Are the limitations of analysis clearly described?

-Do the authors discuss how these data can be helpful to advance our understanding of the topic under study?

-Is public health relevance addressed?

Reviewer #1: (No Response)

Reviewer #3: The revised conclusions are well supported by the data presented and directly linked to the findings on prevalence and intensity across risk groups. The limitations are now more clearly described, including diagnostic sensitivity, intra-county heterogeneity, and loss of data from one cluster, addressing earlier concerns about transparency in interpretation. Still, the limitations section could emphasize more explicitly how diagnostic sensitivity and intra-county heterogeneity may affect programmatic interpretation. The discussion has also been expanded to highlight how these findings contribute to understanding persistent transmission despite long-term school-based deworming, and to situate the results within broader regional and global evidence. The authors now explicitly connect the data to programmatic decisions, including the potential expansion of preventive chemotherapy to PSAC and WRA, and acknowledge alternative treatment strategies for T. trichiura.

The discussion on operational feasibility of expanding PC could also benefit from one or two additional sentences on delivery platforms and cost implications. Public health relevance is clearly articulated, emphasizing the need for tailored interventions, integration with WASH, and alignment with WHO 2030 targets.

The conclusions are stronger, more comprehensive, and clearly address both scientific and programmatic implications, resolving the gaps noted in earlier reviews.

**Editorial and Data Presentation Modifications?**

Reviewer #1: (No Response)

Reviewer #3: The authors have addressed all major concerns raised in previous reviews, including sample size justification, contextualization of the WHO 2% threshold, diagnostic limitations, and expansion of the discussion on reinfection, WASH, and treatment options. Remaining issues are minor:

• While subgroup comparative analyses were not conducted, the justification based on ICSPM methodology is acceptable. However, a short statement acknowledging that such analyses could be explored in future work would strengthen the manuscript.

• The discussion on operational feasibility of expanding PC could benefit from one or two additional sentences on delivery platforms and cost implications, though this is not essential for acceptance.

• The limitations section could emphasize more explicitly how diagnostic sensitivity and intra-county heterogeneity may affect programmatic interpretation.

**Summary and General Comments**

Reviewer #1: (No Response)

Reviewer #3: (No Response)

PLOS authors have the option to publish the peer review history of their article (what does this mean? ). If published, this will include your full peer review and any attached files.

**Do you want your identity to be public for this peer review?** For information about this choice, including consent withdrawal, please see our Privacy Policy .

Reviewer #1: **Yes: ** Gaoussou Coulibaly, Université Félix Houphouët-Boigny, Abidjan, Côte d'Ivoire

Reviewer #3: No

**Figure resubmission:**

**Reproducibility:** To enhance the reproducibility of your results, we recommend that authors of applicable studies deposit laboratory protocols in protocols.io, where a protocol can be assigned its own identifier (DOI) such that it can be cited independently in the future. Additionally, PLOS ONE offers an option to publish peer-reviewed clinical study protocols. Read more information on sharing protocols at https://plos.org/protocols?utm_medium=editorial-email&utm_source=authorletters&utm_campaign=protocols

---

## [Editor Report · Decision Letter 3]

5 Dec 2025

Dear Dr Kibati,

We are pleased to inform you that your manuscript 'Prevalence and intensity of soil-transmitted helminth infections in Narok and Bomet Counties, Kenya: Evidence from program monitoring' has been provisionally accepted for publication in PLOS Neglected Tropical Diseases.

Best regards,

Jong-Yil Chai

Section Editor

Jong-Yil Chai

Section Editor

Shaden Kamhawi

co-Editor-in-Chief

Paul Brindley

co-Editor-in-Chief

This manuscript is now acceptable by the journal. Thank you for your cooperation.

---

## [Editor Report · Acceptance letter]

Dear Dr Kibati,

We are delighted to inform you that your manuscript, "Prevalence and intensity of soil-transmitted helminth infections in Narok and Bomet Counties, Kenya: Evidence from program monitoring," has been formally accepted for publication in PLOS Neglected Tropical Diseases.

Best regards,

Shaden Kamhawi

co-Editor-in-Chief

Paul Brindley

co-Editor-in-Chief
